# Drone Detection and Pose Estimation Using Relational Graph Networks

**DOI:** 10.3390/s19061479

**Published:** 2019-03-26

**Authors:** Ren Jin, Jiaqi Jiang, Yuhua Qi, Defu Lin, Tao Song

**Affiliations:** 1Beijing Key Laboratory of UAV Autonomous Control, Beijing Institute of Technology, Beijing 100081, China; renjin@bit.edu.cn (R.J.); lindf@bit.edu.cn (D.L.); 2Multi-UAV GNC Laboratory, School of Aerospace Engineering, Beijing Institute of Technology, Beijing 100081, China; jiaqi_jiang@bit.edu.cn (J.J.); 3120140024@bit.edu.cn (Y.Q.)

**Keywords:** drone detection, pose estimation, acceleration estimation, relational graph

## Abstract

With the upsurge in use of Unmanned Aerial Vehicles (UAVs), drone detection and pose estimation by using optical sensors becomes an important research subject in cooperative flight and low-altitude security. The existing technology only obtains the position of the target UAV based on object detection methods. To achieve better adaptability and enhanced cooperative performance, the attitude information of the target drone becomes a key message to understand its state and intention, e.g., the acceleration of quadrotors. At present, most of the object 6D pose estimation algorithms depend on accurate pose annotation or a 3D target model, which costs a lot of human resource and is difficult to apply to non-cooperative targets. To overcome these problems, a quadrotor 6D pose estimation algorithm was proposed in this paper. It was based on keypoints detection (only need keypoints annotation), relational graph network and perspective-n-point (PnP) algorithm, which achieves state-of-the-art performance both in simulation and real scenario. In addition, the inference ability of our relational graph network to the keypoints of four motors was also evaluated. The accuracy and speed were improved significantly compared with the state-of-the-art keypoints detection algorithm.

## 1. Introduction

Today, consumer class Unmanned Aerial Vehicles (UAVs) occupy the sky and many applications have emerged [1,2]. Drone detection has become an important issue for low-altitude airspace safety, regulation, and vision-based swarm [3]. Although communication-based models can cope with such problems easily, not all UAVs are equipped with it. Therefore, the ability to use inexpensive optical sensors, such as cameras, to regulate low-altitude airspace [4], avoid collisions [5,6], and search and track target UAVs [7] is becoming increasingly important.

During the past few years, much research has been done on drone detection or small moving object detection [4,5,8,9]. Dey et al. [8] used cascade method to detect aircraft within five miles from 220 degrees of view under Visual Flight Rules (VFR) of U.S. National Airspace (NAS), however, it is mainly used for scenes above the horizon. Rozantsev et al. [5] combined appearance with motion cues, which can detect UAVs and aircraft that occupy only a small part in the field of view, accompanied by the ability to tackle complex backgrounds. Aker et al. [4] proposed a solution using an end-to-end drone detection model based on Convolutional Neural Networks (CNNs). Yoshihashi et al. [9] proposed a method that performed state-of-the-art detection and tracking of small moving objects. However, these methods extract the appearance and motion features to detect the small flying drone with a distance apart, which is mainly used for monitoring and avoidance. Meanwhile, in our case, we hoped to obtain the 6D pose of the drones and make full use of it to understand a UAV’s flight intention. In this way, it could be tracked or efficiently evaded. Besides, pose estimation is also valuable for many applications, such as cooperative control, target motion capture, and behavior analysis.

Most previous studies on target UAVs’ pose estimation algorithms are based on onboard artificial markers. Hajri [10] put six red markers on the same plane of the UAV to estimate its pose. In the experiment, they randomly generated six points on the same plane of simulation environment, and then verified the accuracy of PnP algorithm. The method proposed by Xie et al. [11] required more than three special markers to estimate the UAV’s pose, and these markers could be placed arbitrarily. Fu et al. [12] proposed an off-board quadrotor pose estimation method. They installed four LEDs on the quadrotor and used infrared cameras to estimate its pose. Su et al. [13] also used infrared cameras to detect LEDs mounted on the quadrotor for pose estimation. All the above methods require special identification on the drone. To the best of our knowledge, this is the first work on drone pose estimation without any artificial markers.

At present, there are two main ways to estimate object pose from a single image. One solution is using orientation learning to directly learn the object pose, while the other solution is first to detect a sparse set of category-specific keypoints, and then use such points within a geometric reasoning framework (e.g., PnP algorithm) to recover the object pose. Orientation learning from a single RGB image is a difficult problem because the space of orientations is non-Euclidean [14]. In some traditional studies, attitude estimation is achieved by matching local features from RGB images [14,15,16,17,18]. However, these methods are not suitable for textureless objects. Recently, most studies use machine learning, especially deep learning, to estimate the 6D pose of objects or a camera, like Reference [19,20,21,22,23,24,25]. Kehl et al. [19] proposed an extension of Single Shot MultiBox Detector (SSD) [26] that produces 2D detections and infers proper 6D poses, where its input in training stage is the synthetic 3D model information. Mahendran et al. [20] tried to directly regress full 3D object orientation by natural non-linearity output layer and an appropriate geodesic loss, and they used fine labeled 3D object with pose PASCAL3D+ [27]. Rad et al. [21] introduced the classifier before estimating the pose to identify the attitude range at runtime. In the stage of pose estimation, the method from Crivellaro et al. [28] was used. They also introduced a CAD model to produce 2D projection prior, to improve the accuracy of control points. However, these methods all depend on CAD model or accurate 3D pose annotation, and the information is difficult to obtain in practice for non-cooperative targets.

2D keypoints detection is a long-standing research problem in computer vision [29] and it is traditionally used as an early stage in the object localization algorithm [30]. 2D human joint keypoints detection from monocular RGB images is a successful early application of modern CNNs. Due to its compelling utility for human computer interaction (HCI), motion capture, and security applications, a large number of work has since developed in this human keypoints detection domain [31,32,33,34,35,36,37]. The definition of human keypoints is in the image coordinate system, but for quadrotors, the correct order of four motors should be located. Our experiments indicated that directly using the 2D human keypoints detection algorithm could not achieve better results, as shown in Section 3.3.

In the real world, a variety of data exhibit much richer relational graph structures than the simple grid-like. For example, in the field of language, linguists use parsing trees to represent the syntactic dependence between words, and information retrieval systems use knowledge graphs to reflect entity relations. In the domain of vision, modeling the relations between pixels has also been proven useful [38,39,40,41].

Based on the above observation, we proposed a novel relational graph networks to inference quadrotor’s keypoints, which can improve the average precision by 9.7 percent compared with baseline method in our Parrot quadrotor keypoints dataset. By taking the quadrotor as our main study object, four motor axes were first detected as four keypoints. Its order was derived from relational neural networks to reduce the dependence on CAD models. After resolving the direction of *Z* axis to increase stability, the improved PnP algorithm was utilized to accurately estimate the 6D pose of quadrotor. Compared with the methods mentioned above, our detection and keypoints estimation were trained by end to end, and only the keypoints annotation was required, which was easier to get. Besides, we estimated the acceleration of the quadrotor using the solved 6D pose. Experiments showed that the state tracking error could be greatly reduced.

The main contributions of this paper are: (1) We proposed a keypoints detection network with relational graph, which could effectively improve the keypoints detection accuracy and speed of the quadrotor. (2) Combined with keypoints detection results, the improved PnP algorithm and the filtering algorithm, we demonstrated that the method could actually reduce the UAV’s state tracking error.

## 2. 6D Drone Pose Estimation

In order to estimate the 6D pose of the target quadrotor, we chose the way to locate the keypoints of the four motors, as shown in Figure 1. This way could decrease the labeling effort to the maximum extent and could use real images to reduce the cost of domain adaptation. Since the order of four motors in the image needed to be determined according to the direction of the azimuth of the quadrotor and the viewing angle (looking down or looking up) by the observation camera, we annotated eight keypoints on the body of target quadrotor, named as motor 1, motor 2, motor 3, motor 4, nose, tail, body top, and body bottom. The keypoints that do not appear in the image or are obscured would also be marked.

### 2.1. Relational Keypoints

During the learning phase, the eight keypoints were divided into two groups. The keypoints in first group had obvious appearance characteristics, such as nose, tail, body top, and body bottom, which are defined as anchor keypoints. The keypoints in the second group had a similar appearance and with a logical distinction, like motor 1, motor 2, motor 3, and motor 4, which we titled relational keypoints.

**Anchor Keypoints Predictor**: Due to the fact that keypoints detection is based on the quadrotor detection results, similar to Mask R-CNN [42], all of them can be carried out simultaneously by a two stages detector. The first stage is a Region Proposal Network (RPN) [43], which is used for proposing candidate object bounding boxes. In the second stage, the class, bounding box offset and keypoints are predicted in parallel. Define $R^{0}$ as the output of RoIAlign [42]. The anchor keypoints predictor is defined as:(1)$$A^{l}=ReLU\left(A^{l-1}⊗W_{a}^{l}+b_{a}^{l}\right),$$
where $A^{l-1}$ and $A^{l}$ are the outputs of l−1 and *l* layer, respectively, and $A^{0}=R^{0}$ is the default input. $W_{a}^{l}$ and $b_{a}^{l}$ are model parameters, and the bias at layer *l*, ⊗ represents convolution operation.

**Relational Keypoints Predictor**: The relational keypoints predictor is divided into two parts. The first part is a graph predictor, which is used to encode relevant information about anchor keypoints and relational keypoints. Given the input RoI $R^{0}$, set $A^{l}$ defined above as a key CNN, $Q^{l}$ as a query CNN, which has similar structure and outputs with $A^{l}$. We define the graph predictor at layer *l* as:(2)$$G_{ij}^{l}=\frac{{\left(ReLU\left(a_{i}^{l⊤}q_{j}^{l}+b_{g}^{l}\right)\right)}^{2}}{\sum_{i^{\prime }}{\left(ReLU\left(a_{i^{\prime }}^{l⊤}q_{j}^{l}+b_{g}^{l}\right)\right)}^{2}},$$
where $a_{i}^{l}=W_{a}^{l}A^{l}$ and $q_{i}^{l}=W_{q}^{l}Q^{l}$. $W_{a}^{l}$ and $W_{q}^{l}$ are model parameters matrices at layer *l*, and $b_{g}^{l}$ is a scalar bias parameter. This is similar to the non-local neural network [41] and the attention model [44], however, the difference is we used anchor keypoints predictor as the input of attention. The ReLU operation was used to enforce sparsity and the square operation to stabilize training. In addition, we added stacked convolutional networks to allow the graph predictor to be aware of the local order of the context and also to increase receptive field of the networks.

The second part of relational keypoints predictor is the feature predictor. Let the features $F^{0}=R^{0}$ be an input layer. We first added the convolutional layer to extract features and increase receptive field. The affinity matrix $G^{l}$ was then combined with the current features to produce the next layer’s features:(3)$$F^{l}=\sum_{j}G_{j}^{l}F_{j}^{l-1}+F^{l-1}.$$

The feature at each position was calculated as the weighted sum of other features, where the weights were determined by graph $G^{l}$, followed by residual connections.

**Objective Function**: We obtained the features $A^{L}$ and $F^{L}$ at the top layer of anchor keypoints predictor and relational keypoints predictor, respectively. The objective of the anchor and relational keypoints predictor are written as:(4)$$max\sum_{i}\sum_{k}x_{ik}\log P\left(x_{ik}|A^{L}\right)+max\sum_{i}\sum_{k}x_{ik}\log P\left(x_{ik}|F^{L}\right),$$
where *i* represents the spatial index in the final layer $A^{L}$/$F^{L}$, and *k* denotes the class of anchor/relational keypoints. The overall keypoints objective function is the sum of the above two functions.

### 2.2. Detection Framework

So far, there exist two kinds of CNN-based object detectors. One is the single-stage detector mentioned in Reference [26,45,46,47], which has the advantage of a very fast speed and reasonably good accuracy. The other is the two-stage detector described in Reference [42,43,48,49], where the first stage (body) uses the region proposal network to generate many proposals, and the second stage (head) is to recognize these proposals. The advantages of the two-stage detector are its precision and scalability. It can add more than one head to accomplish many tasks at the same time, such as mask and keypoints detection [42]. The disadvantage is that by using heavy head to get great accuracy, it cannot run at real time, even on a desktop with a Titan GPU. Inspired by Li et al. [50] and He et al. [42], we designed a light head two-stage detector that detects drone and its keypoints simultaneously. It could reach 71 fps on our desktop computer. The complete detection framework is shown in Figure 2.

**Body Network and Thin Features**: We used the Xception-like network model as the base network for feature extraction and ensuring real time computational speed. The network structure of the Xception model is shown in Table 1. The large separable convolution layers [51] were added on conv5 of Xception. This not only could effectively compress features and improve network performance, but also could get more powerful feature maps from a larger receptive field with large kernels [50]. In our experiments, large separable convolution with kernel size = 15, $C_{mid}=64$ and $C_{out}=128$ was applied on conv5 to obtain light head feature maps.

**Head networks**: For object classification and localization, we applied a single fully connected layer with 512 channel; after that, two parallel fully connected layers were used to predict RoI classification and regression. Four channels were deployed for each bounding box location because the regression was shared between different classes. For keypoints detection, two sets of convolution layers were followed by Region of Interest (RoIs), one for predicting anchor keypoints and the other for predicting relational keypoints. The relational keypoints used the feature maps of anchor keypoints as an input to establish the relevant model. In our experiments, RoIAlign was used to generate RoI boxes, and pooler resolution was set to 14 to improve the localization accuracy of the keypoints. In a minibatch, the fraction of foreground RoI was assigned to 0.25 and batch size was set to 512 in one image. The specific description is in Section 3.1, and its network structure is shown in Figure 2.

**RPN**: RPN is a class agnostic object detector; it pre-defines a set of anchors, which are controlled by several specific scales and aspect ratios. The anchor will be set as a positive label if it has intersection over union (IoU) over 0.7 with any ground-truth box. Anchors which have the highest IoU for the ground-truth box will also be given a positive label. Meanwhile, if extra anchors have IoU less than 0.3 with all ground-truth boxes, their labels will be assigned as negative [43]. In our experiments, we built RPN on conv4 of the Xception backbone. Two aspect ratios {1:1,2:1} and five scales $\left\{32^{2},64^{2},128^{2},256^{2},512^{2}\right\}$ anchors were used to cover prior drone shape and possible different sizes. The anchor stride was set to 16 and the fraction of foreground (positive) examples in each image batch was 0.5.

**Training Loss**: We used a multi-task loss function for each RoI during training time:(5)$$\begin{array}{cc} L(c_{dt},c_{gt},t_{dt}^{c},t_{gt},k_{dt}^{c},k_{gt}) & =L_{cls}\left(c_{dt},c_{gt}\right)+\\ & \alpha \left[c_{gt}\geq 1\right]L_{loc}\left(t_{dt}^{c},t_{gt}\right)+\\ & \beta \left[c_{gt}\geq 1\right]L_{kp}\left(k_{dt}^{c},k_{gt}\right), \end{array}$$
where $L_{cls}\left(c_{dt},c_{gt}\right)=-logp_{c}$ is the softmax loss for true class *c*. $L_{loc}\left(t_{dt}^{c},t_{gt}\right)=smooth_{L1}\left(t_{dt}^{c}-t_{gt}\right)$ is the smooth L1 loss proposed in Reference [48], where $t_{gt}$ is the true bounding-box regression target. $L_{kp}\left(k_{dt}^{c},k_{gt}\right)$ is the pixel softmax loss function of each keypoint, which is divided into two parts (the details are introduced in Section 2.1). $\left[c_{gt}\geq 1\right]$ is an indicator function, andd the output value will be set to 1 if $c_{gt}\geq 1$, otherwise 0. α and β are balance parameters, which we set to 1 in our experiments.

### 2.3. PnP Pose Estimation

Accurate pose estimation relies on keypoints, besides, an algorithm is needed to solve the attitude from these keypoints. Furthermore, the estimated pose can be utilized to calculate the acceleration of quadrotor. Since the labeling errors of the head, tail, top, and bottom are large, the order and position of the four motors are relatively accurate. We only used these four points to solve the relative pose of the target quadrotors. The real quadrotor attitude in the world frame could be decoupled according to the pose of the camera.

Due to the influence of annotation errors and keypoints detection noise, the PnP algorithm was unstable for plane solution. In the same experimental environment, the *Z* axis of the previous frame was vertical upward, and the latter frame was downward, as shown in Figure 3a,b. And the *Z* axis we defined is upward, so the decoupled attitude was totally wrong. Our improved PnP solved all the possible *R* and *t* with current observations, and then used a test point to determine the correct estimates of *R* and *t*. First, let us review the solution of all the possible *R* and *t*. The detailed algorithm can be found in Reference [52].

Given the 3D position of the motor in drone body frame and the position of the keypoints in the image, the solution of its relative attitude constitutes a perspective-n-point (PnP) problem. Consider three reference points $P_{1}$, $P_{2}$ and $P_{3}$, and three constraints by dividing them into 2-points are obtained:(6)$$\left\{\begin{array}{l} x_{1}^{2}+x_{2}^{2}-2x_{1}x_{2}\cos\theta_{12}-d_{12}^{2}=0\\ x_{1}^{2}+x_{3}^{2}-2x_{1}x_{3}\cos\theta_{13}-d_{13}^{2}=0\\ x_{2}^{2}+x_{3}^{2}-2x_{2}x_{3}\cos\theta_{23}-d_{23}^{2}=0, \end{array}\right.$$
where $x_{1}$, $x_{2}$, and $x_{3}$ are unknown depths from the reference points to the camera center and $d_{12}$, $d_{13}$, and $d_{23}$ are the known distance between $P_{1}P_{2}$, $P_{1}P_{3}$, and $P_{2}P_{3}$, respectively. $\theta_{12}$, $\theta_{13}$, and $\theta_{23}$ are the viewing angles from the camera center to $P_{1}P_{2}$, $P_{1}P_{3}$, and $P_{2}P_{3}$, respectively (see Figure 4). The equation system can be converted into a fourth order polynomial equivalently with three unknown depth variables $x_{1}$, $x_{2}$, and $x_{3}$ [52,53]:(7)$$f\left(x\right)=ax^{4}+bx^{3}+cx^{2}+dx+e=0.$$

For four points, we can get four group of combinations of three points. In order to solve these polynomials, a cost function is defined as $F=\sum_{i=1}^{4}f_{i}^{2}\left(x\right)$. The minima of *F* can be determined by finding the roots of its derivative $F^{\prime }=\sum_{i=1}^{4}f_{i}\left(x\right)f_{i}^{\prime }\left(x\right)=0$. *F* has at most four minima, and the proof can be found in Reference [52].

Since these four points are in the same plane, the direction of the *Z* axis in the body coordinate system could not be well determined. Under the same experimental conditions, the direction of the *Z* axis was ambiguous in the two frames (see Figure 3a,b).

For each minimum, the rotation matrix $R_{i}$ and translation matrix $t_{i}$ from quadrotor body frame to camera frame and their $loss_{i}$ could be solved by Reference [52]. Then, a test case is added to solve the problem of ambiguity. Assume the *Z* axis of the aircraft system is perpendicular to the quadrotor plane. The algorithm is as follows, Algorithm 1:
**Algorithm 1:** Choosing the most suitable solution from possible solutions.
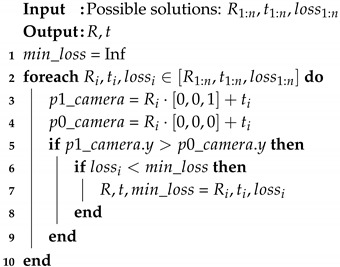


Then we solved z-axis problem based on the quadrotor body system (the result is shown in Figure 3c,d).

## 3. Experiments

In this part, we designed experiments to evaluate our algorithm. The experiments were divided into three parts. In the first part, we used simulation data to evaluate the keypoints detection algorithm, and mainly verified the performance of the relational graph keypoints head. In the second part, the drone Parrot Bebop-2 [54] was used to produce two real datasets in an indoor and outdoor environment, respectively, covering the situation of looking down and looking up, as well as different perspectives. On these datasets, we validated the effectiveness of our keypoints detection algorithm and the important role of the proposed relational graph networks. In the last part, the improved PnP algorithm was used to solve the 6D pose information of quadrotor and its value was compared with ground truth. From the 6D pose, the acceleration in drone body frame could be obtained, and we compared the velocity tracking performance with and without acceleration information by using a Kalman filter method.

### 3.1. Implementation Details

The Xception-like model was used as our backbone network (its details are shown in Table 1). The output channels of conv4 and conv5 were 576 and 1152, respectively. Large separable convolution with kernel size = 15, $C_{mid}=64$ and $C_{out}=128$ was applied on conv5 to obtain light head feature maps. In the implementation of RPN, we used two aspect ratios {1:1,2:1} and five scales $\left\{32^{2},64^{2},128^{2},256^{2},512^{2}\right\}$ anchors to cover prior drone shapes and possible different sizes. A stack of eight convolutional layers was used for predicting the anchor keypoints as one-hot masks. For relational keypoints detection, we inserted relational graph layers into the relational keypoints head after every two convolutional layers and anchor keypoints features of corresponding layer were used as its input.

The whole detector was end-to-end trained based on four Nvidia Tesla V100 GPUs using synchronized Stochastic Gradient Descent (SGD) with a momentum of 0.9 and a weight decay of 1e-4. Each mini-batch had two images per GPU, and each image had 200/100 RoIs for training/testing. For the simulation dataset, the learning rate was set to 0.01 for first 200 K iterations (passing one image would be regarded as one iteration) and 0.001 for later 65K iterations. For the real dataset, the number of iterations was 400K and 130K for the 0.01 and 0.001 learning rate, respectively. The backbone network was initialized based on the pre-trained ImageNet [55] base model, and we fixed batch normalization for faster training. Online hard example mining [56] was also used in our experiments.

In the following section, we will introduce three parts of experiments in detail: the keypoints detection on quadrotor simulation dataset, the keypoints detection on real dataset of Parrot Bebop-2, and the comparative experiment of 6D pose analysis.

### 3.2. Keypoints Detection on Simulation Dataset

The simulation data had more consistent image characteristics, which were exactly the same for the four motors. The quadrotor’s nose and tail were represented by special shapes. Using such simulation dataset could clearly evaluate the reasoning ability of the network for the four motors of the quadrotor. Some examples are shown in Figure 5.

An X-type quadrotor model was built on the same plane. We used a circle to represent one motor axis, and four motor axes were expressed in the same way. The triangle represented the nose, and the horizontal line represented the tail. After that, a simulated pinhole camera model was set up to observe the quadrotor.

Let the simulation quadrotor’s yaw rotate in the range of ± 180 degrees, roll in the range of ±45 degrees, and pitch in the range of 0 to 45 degrees. In this way, the 6D pose could be observed within the normal working range. We generated 6804 simulation images of different pose; only 5 percent (340) of these were randomly selected as the training dataset and the rest (6464) as the testing dataset. It was shown that the training data did not cover all possible poses, and we wanted the network to learn to reason from a small amount of data and get the relationship between the keypoints of four motors, nose, and tail.

Only four keypoints of the rotors were evaluated, which contributed to the final quadrotor’s pose estimation. The other keypoints which depend on the specific quadrotors without containing enough common feauters could be used to help locate the keypoints of the motors. In order to evaluate the performance of keypoints detection, we used an evaluation method similar to COCO dataset [57]. The object keypoint similarity (*OKS*) was defined as:(8)$$OKS=\frac{\sum_{i}\exp\left(-d_{i}^{2}/2s^{2}k_{i}^{2}\right)\delta \left(v_{i}>0\right)}{\sum_{i}\delta \left(v_{i}>0\right)},$$
where $d_{i}$ is Euclidean distance between each detected keypoint and its corresponding ground truth values. $v_{i}$ is the visibility flag of the ground truth, and the prediction $v_{i}$ of the detector is not used. *s* is the object scale. We adjusted the $k_{i}$ to make the *OKS* a perceptive and easy to interpret similarity measure. The redundant annotations of the real dataset were used to calculate the standard deviation of the keypoints, $\sigma_{i}^{2}=E\left(d_{i}^{2}/s^{2}\right)$, $k_{i}=2\sigma_{i}$ and $\sigma_{i}$ is the variance of manual annotation for keypoint *i*. We set $k_{i}=0.15$ according to redundant manual annotation of the real data.

Next, true positive (*TP*) is expressed as the number of *OKS* larger than the threshold, false positive (*FP*) is the sum of keypoints number that *OKS* smaller than the threshold and the ground truth keypoints number of all missing objects, false negative (*FN*) is the number of all false keypoints detections (including the keypoints of false object detections). The main evaluation criteria average precision (*AP*) and average recall (*AR*) are as follows:(9)Precision=TP/(TP+FN),Recall=TP/(TP+FP).

We set the threshold as 0.5, 0.75, and 0.5:0.05:0.95 (take the average of these 10) to define $AP_{OKS=0.5}$, $AP_{OKS=0.75}$, and $AP_{OKS=0.5:0.05:0.95}$, respectively. Similarly, three average recall criteria were defined as $AR_{OKS=0.5}$, $AR_{OKS=0.75}$, and $AR_{OKS=0.5:0.05:0.95}$. The evaluation results of the simulation quadrotor dataset are shown in Table 2, where *Lh-rcnn-k4* indicates that only four keypoints (motor 1, motor 2, motor 3, and motor 4) were used and keypoints head I was applied. *Lh-rcnn-k8* means eight keypoints (nose, tail, body top, and body bottom are added) and also head I are used. *Lh-rcnn-k4-4* represents eight keypoints were split into two groups (motor 1, motor 2, motor 3, and motor 4 are called relational keypoints; nose, tail, top and bottom are called anchor keypoints) and head II was deployed. *Lh-rcnn-k8-NL* used one group eight keypoints with non-local head I, and *Lh-rcnn-k4-4-RG* used two group keypoints and head III, and the suffix NL and RG denote the non-local block and relational graph block described above, respectively.

From the experimental results, it could be seen that the keypoints detection performance was significantly improved by dividing the keypoints into anchor and relational. And after adding head III with the relational graph, the state of the art result was obtained (see Table 2).

### 3.3. Keypoints Detection on Parrot Dataset

We collected and annotated the keypoints of real data to evaluate the algorithm. Parrot Bebop-2 is a kind of consumer-grade quadrotor, and its four motors have similar characteristics. The sequence needs to be inferred according to the nose, tail, and perspective, as shown in Figure 1.

The first dataset was collected in an indoor environment. In this way, a motion capture system (OptiTrack [58]) could be used to compare the results of 6D pose tracking. The dataset contains the perspective of looking down and looking up at the quadrotor Parrot. Then we captured images from multiple azimuth observation perspectives, some of which were used as training sets (3957 images), and others were used as evaluation sets (1670 images). In addition, to verify the adaptability of the algorithm in the more complex scenario, we captured 3189 images in three outdoor scenes, and split them into two parts randomly: one part for training (2232 images) and the other for testing (957 images). These images are artificially labeled with the eight keypoints described above. Some examples of annotations are shown in Figure 6. These two datasets are named Parrot indoor dataset and Parrot outdoor dataset, respectively.

Similar to the evaluation method used in simulation data, we only evaluated four motors’ keypoints, which were used to calculate 6D pose of the quadrotor in the next step. $AP_{OKS=0.5}$, $AP_{OKS=0.75}$, $AP_{OKS=0.5:0.05:0.95}$
$AR_{OKS=0.5}$, $AR_{OKS=0.75}$, and $AR_{OKS=0.5:0.05:0.95}$ were also used as the keypoints evaluation criteria. The evaluation results on the Parrot indoor dataset are shown in Table 3.

From the above comparison experiments, we could see that the relational graph network still effectively improved the detection accuracy of keypoints in the real dataset. *Lh-rcnn-k4-4-RG* indicated that eight keypoints were split into two groups; motor 1–4 as group 1 (motor 1, motor 2, motor 3, and motor 4, called relational keypoints) and the other four keypoints as group 2 (nose, tail, body top, and body bottom, called anchor keypoints). RG means the relational group networks were used in the detection framework, as shown in Figure 2 III. However, the overall accuracy of keypoints detection in real data was lower than the experimental results of simulation data. This was because the precision of manual annotation brought more noise to the dataset, and the real data had the influence of motion blurring and complex background. Our experiments on the more complex outdoor dataset also confirmed this observation. The evaluation results on Parrot outdoor dataset are shown in Table 4.

Then we compared the running speed of these models on a desktop computer with intel i7 6700K CPU and Nvidia Titan X GPU (the results are shown in Table 5), and we could see that *Lh-rcnn-k4-4-RG* had advantages in both accuracy and speed.

After getting the keypoints of the quadrotor, we needed to evaluate the results of the 6D pose estimation. For this purpose, a one-minute test set was captured; meanwhile, the real-time 6D pose of the camera and the target quadrotor were recorded by the motion capture system as our ground-truth. *Lh-rcnn-k4-4-RG* keypoints detection framework and improved PnP algorithm were used to generate the 6D pose of the quadrotor. The comparison results of its position and attitude are shown in Figure 7 and Figure 8, respectively.

It could be seen that besides the noise caused by the keypoints detection error, the 6D pose tracking results were accurate. These noises could be smoothed by Kalman filter in the subsequent processing. In the following part, we used the 6D pose of the quadrotor to estimate its acceleration and combined the Kalman filter with acceleration estimation to get more accurate target velocity estimation.

### 3.4. Experiments on State Estimation

In order to describe the motion of the quadrotor, the Parrot Bebop-2 in Figure 1 is represented by a rigid body of mass *m*. Meanwhile, a world inertial frame $\sum_{w}$ and a body-fixed frame $\sum_{b}$ attached to the quadrotor at the mass center are introduced as a two reference frame. The position, linear velocity, and attitude (roll/pitch/yaw angles) of the quadrotor are represented as $p={\left[x,y,z\right]}^{⊤}$, $v={\left[\dot{x},\dot{y},\dot{z}\right]}^{⊤}$ and $\phi ={\left[\varphi ,\theta ,\psi \right]}^{⊤}$, respectively. The translation model of the quadrotor is as follows:(10)$$\ddot{x}=\frac{F}{m}\left(\sin\theta \cos\psi +\sin\varphi \cos\theta \sin\psi \right),$$
(11)$$\ddot{y}=\frac{F}{m}\left(\sin\theta \sin\psi -\sin\varphi \cos\theta \cos\psi \right),$$
(12)$$\ddot{z}=\frac{F}{m}\left(\cos\varphi \cos\theta -g\right),$$
where *g* is the gravity constant, and $F=F_{1}+F_{2}+F_{3}+F_{4}$ represents the total thrust of four motors. The target quadrotor is assumed to be in a horizontal stable flight state, i.e., $a_{z}=0$. With Equations (10)–(12), we can calculate the acceleration $a={\left[a_{x},a_{y},a_{z}\right]}^{⊤}={\left[\ddot{x},\ddot{y},\ddot{z}\right]}^{⊤}$ by quadrotor’s attitude:(13)$$a_{x}=\frac{g}{\cos\varphi \cos\theta }\left(\sin\theta \cos\psi +\sin\varphi \cos\theta \sin\psi \right)$$
(14)$$a_{y}=\frac{g}{\cos\varphi \cos\theta }\left(\sin\theta \sin\psi -\sin\varphi \cos\theta \cos\psi \right).$$

In the previous section, the position and attitude of the quadrotor were calculated in the camera frame. Here, we need to get the position and attitude of the quadrotor in the world frame, according to the camera attitude obtained by OptiTrack. In the real scene, the camera’s attitude could be achieved by its own inertial measurement unit (IMU).

Next, a formulation of the Kalman filter is presented to estimate the position and velocity of the quadrotor as follows:(15)$$X_{k}={\mathrm{AX}}_{k-1}+w_{k},$$
(16)$$Z_{k}={\mathrm{HX}}_{k}+v_{k},$$
where $w_{k}$ and $v_{k}$ indicate the system and measurement noises, respectively. Since the position and acceleration of the quadrotor are included in the system measurement Z, we used the near-constant acceleration (NCA) as our model. The state vector, measurement vector, state matrix, and measurement matrix are given as follows: (17)$$X:=\left[\begin{array}{c} p\\ v\\ a \end{array}\right],Z:=\left[\begin{array}{c} p\\ a \end{array}\right],$$
(18)$$A=\left[\begin{array}{ccc} I_{3} & hI_{3} & \frac{1}{2}h^{2}I_{3}\\ 0 & I_{3} & hI_{3}\\ 0 & 0 & I_{3} \end{array}\right],H=\left[\begin{array}{ccc} I_{3} & 0 & 0\\ 0 & 0 & I_{3} \end{array}\right],$$
where *h* denotes the sampling period, and $I_{n}$ indicates dimension identity matrix of *n*-dimension. The covariance matrix of $v_{k}$ is then defined as:(19)$$Q=E\left[v_{k}v_{k}^{\prime }\right]=\left[\begin{array}{ccc} \frac{1}{20}h^{5}I_{3} & \frac{1}{8}h^{4}I_{3} & \frac{1}{6}h^{3}I_{3}\\ \frac{1}{8}h^{4}I_{3} & \frac{1}{3}h^{3}I_{3} & \frac{1}{2}h^{2}I_{3}\\ \frac{1}{6}h^{3}I_{3} & \frac{1}{2}h^{2}I_{3} & hI_{3} \end{array}\right]q,$$
where *q* is the power spectral density of the process noise in this model. The detailed explanation of coefficients and a guideline for the choice of *q* can be found in Reference [60]. The state prediction and the measurement update are given by:(20)$${\hat{X}}_{k|k-1}=A{\hat{X}}_{k-1|k-1}+w_{k-1},$$
(21)$${\hat{X}}_{k|k}={\hat{X}}_{k|k-1}+\gamma_{k}K_{k}\left(Z_{k}-H{\hat{X}}_{k|k-1}\right),$$
where $K_{k}$ represents the observer gain. Due to the dynamic change of illumination conditions, imaging noise, and motion blur, the detection of the target might occasionally be lost. $\gamma_{k}$ is a binary stochastic variable to model the intermittent measurements [61]. If the target is detected and a measurement arrives after the *k* th step, $\gamma_{k}=1$, and if no measurement appears after the *k* th step, then $\gamma_{k}=0$.

To verify the effectiveness of the estimation system, we contrast our estimation results with ground truth measured by OptiTrack system in Figure 9 and Figure 10. Another estimation result based on near-constant velocity (NCV) model is also given for comparison. The mean error and standard deviation of the estimation error are shown in Table 6.

From Figure 9 and Figure 10 and Table 6, it can be seen that the NCA model could track the position and velocity faster than the NCV model. It is workable to calculate the acceleration of the leader quadrotor using the vision-measured attitude, which can improve the estimation result.

Some representative examples in the evaluation dataset and the test dataset are shown in Figure 11. It can be seen that the detection algorithm could accurately locate the keypoints from various perspectives and give the correct attitude calculation.

## 4. Conclusions

In this paper, we proposed a vision-based quadrotor pose and acceleration estimation method. Our method included two parts: one is a novel relational graph network to improve the keypoints detection performance, and the other is an improved PnP algorithm to acquire 6D quadrotor’s pose. These two algorithms were further integrated to estimate the acceleration and accurate velocity of the target. Experiments and ablation studies (shown in Table 3) indicated that our relational graph network enhanced 9.7% average precision compared to the baseline network in the same configuration. Besides, position and velocity errors (shown in Table 6) could be reduced by 19% and 40%, respectively, with the integration of acceleration estimation.

The compatibility and performance of the algorithm was significantly adequate. It could achieve 71 fps on a desktop computer with an Nvidia Titan X GPU. Moreover, the algorithm was independent of the 3D model or accurate pose annotation for 6D pose estimation, meanwhile, it only required the keypoints annotation of the target using captured images and was especially suitable for a non-cooperative target.

The downside of this algorithm was the accuracy of 6D pose decreased with the shrink of object size for small targets in the image. Therefore, in the future, we will improve the results on small target by employing adversarial training methods. 

## Figures and Tables

**Figure 1 sensors-19-01479-f001:**
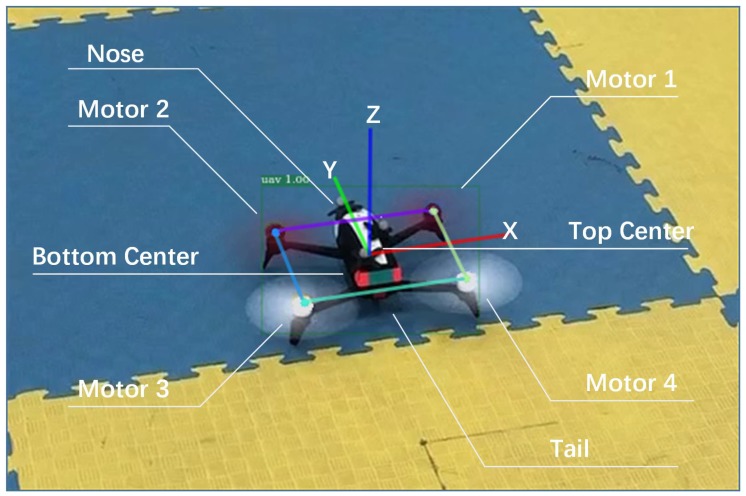
Parrot Bebop-2 quadrotor and the definition of its eight keypoints.

**Figure 2 sensors-19-01479-f002:**
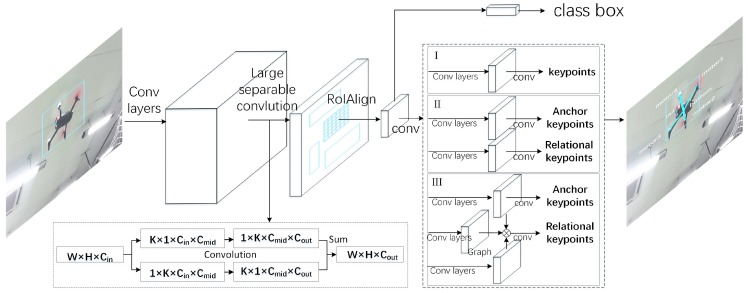
The quadrotor and its keypoints detection framework. We validate three kinds of keypoints heads; see the text description for the details.

**Figure 3 sensors-19-01479-f003:**
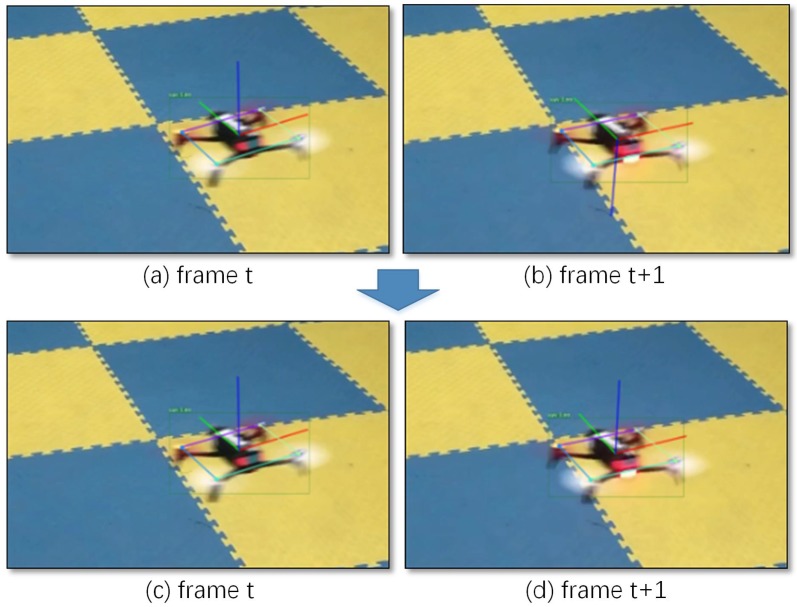
Solve the problem of *Z* axis in PnP. In two adjacent frames, *t* and t+1, under similar circumstances, the *Z* axis of the frame *t* is vertical upward (**a**), and of the frame t+1 is downward (**b**). As the *Z* axis we defined is upward, our improved PnP algorithm can solve the problem by a test case (**c**,**d**); see the text for details.

**Figure 4 sensors-19-01479-f004:**
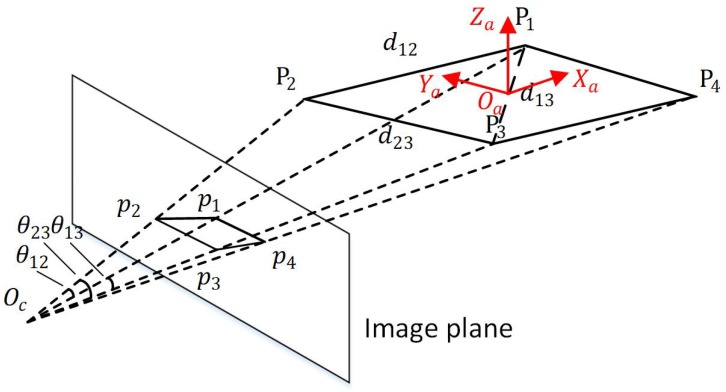
The projection of the reference points and three constraints of 2-points.

**Figure 5 sensors-19-01479-f005:**
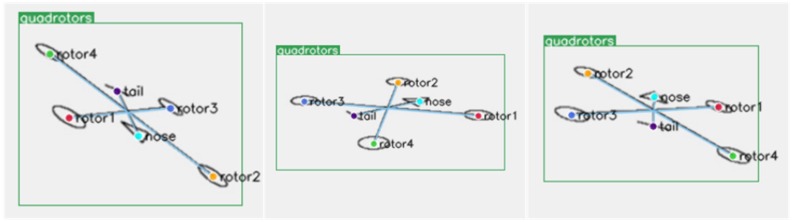
Some examples of quadrotor simulation data. It can be seen that the order of the motor shaft must be deduced from the nose and tail.

**Figure 6 sensors-19-01479-f006:**
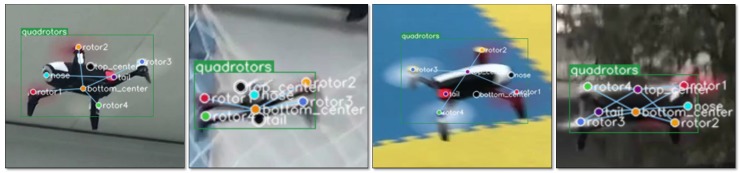
Some keypoints annotation examples of our Parrot dataset.

**Figure 7 sensors-19-01479-f007:**
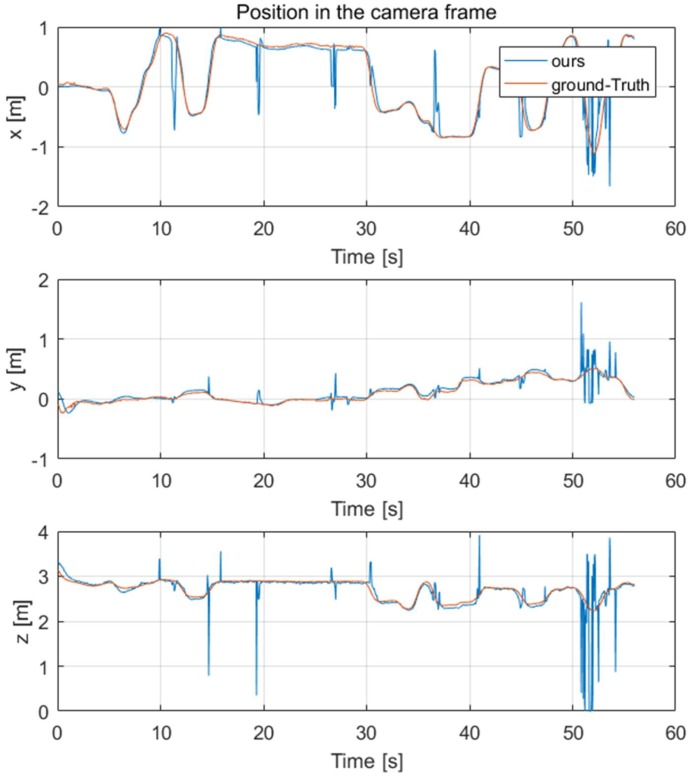
The position estimation in the camera frame, compared with OptiTrack (as Ground-Truth).

**Figure 8 sensors-19-01479-f008:**
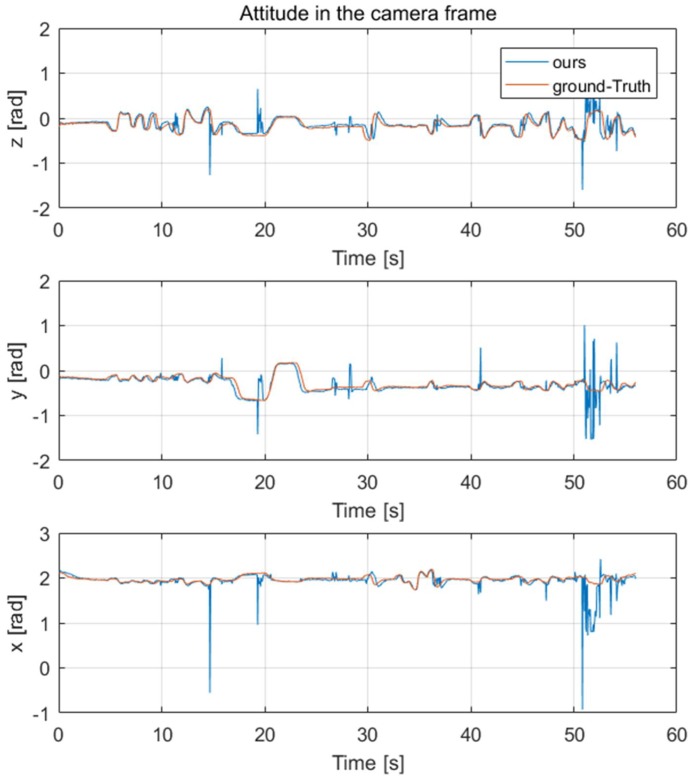
The attitude estimation in the camera frame, compared with OptiTrack (as Ground-Truth).

**Figure 9 sensors-19-01479-f009:**
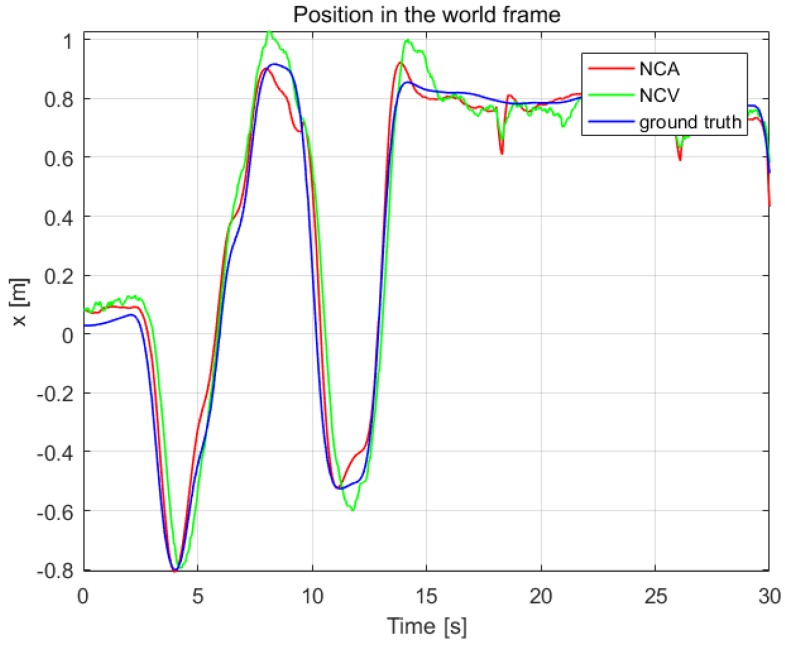
The position estimation in the world frame, compared with OptiTrack.

**Figure 10 sensors-19-01479-f010:**
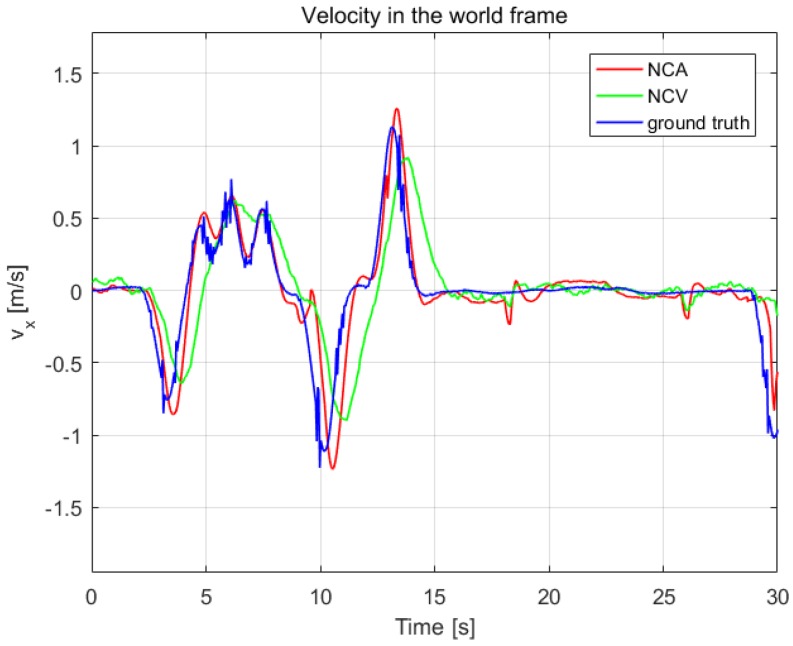
The velocity estimation in the world frame, compared with OptiTrack.

**Figure 11 sensors-19-01479-f011:**
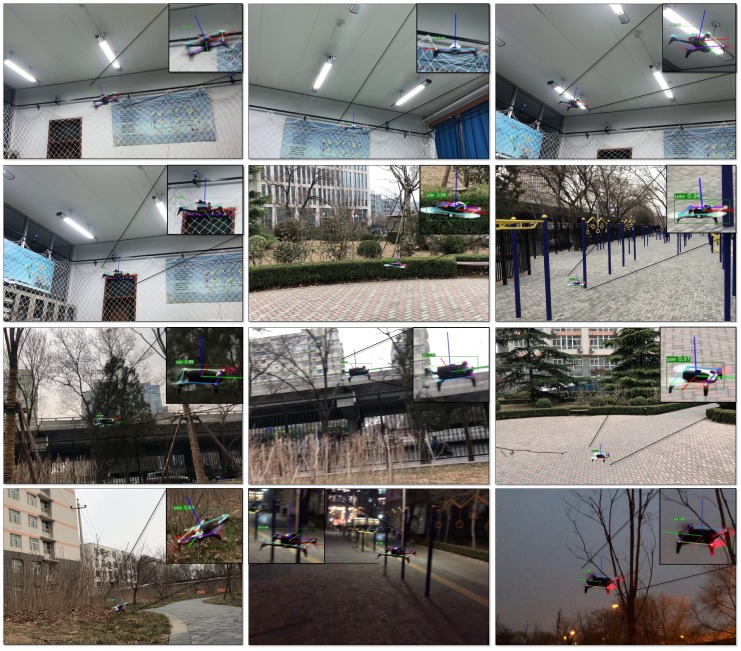
Representative results of quadrotor pose estimation by our relational graph keypoints detection model and PnP algorithm. It can be seen that the quadrotor pose can be estimated from a variety of observation angles.

**Table 1 sensors-19-01479-t001:** The detail architecture of Xception like backbone used in our detection and keypoints network.

Layer	Output Size	Kernel Size	Scale	Repeat	Output Channels
Image	224×224				
Conv1	112×112	3×3	2	1	24
Max Pool	56×56	3×3	2		
Conv2	28×28		2	1	144
	28×28		1	3	144
Conv3	14×14		2	1	288
	14×14		1	7	288
Conv4	7×7		2	1	576
	7×7		1	3	576
GAP	1×1	7×7			576
FC					1000

**Table 2 sensors-19-01479-t002:** Comparison of different keypoints head on simulation quadrotor dataset. (AP and AR represent the average accuracy and average recall, respectively. OKS indicates object keypoint similarity. For the explanations of each model, please refer to the text.)

**Model**	${\mathrm{AP}}_{\mathrm{OKS}=0.5:0.95}$	${\mathrm{AP}}_{\mathrm{OKS}=0.5}$	${\mathrm{AP}}_{\mathrm{OKS}=0.75}$
Lh-rcnn-k4	0.8176	0.8412	0.8001
Lh-rcnn-k8	0.8123	0.8397	0.7985
Lh-rcnn-k8-NL	0.8189	0.8476	0.8091
Lh-rcnn-k4-4	0.9083	0.9346	0.9055
Lh-rcnn-k4-4-RG (ours)	0.9366	0.9437	0.9212
**Model**	${\mathrm{AP}}_{\mathrm{OKS}=0.5:0.95}$	${\mathrm{AP}}_{\mathrm{OKS}=0.5}$	${\mathrm{AP}}_{\mathrm{OKS}=0.75}$
Lh-rcnn-k4	0.8345	0.8551	0.8152
Lh-rcnn-k8	0.8252	0.8435	0.8124
Lh-rcnn-k8-NL	0.8358	0.8576	0.8173
Lh-rcnn-k4-4	0.9107	0.9407	0.9060
Lh-rcnn-k4-4-RG (ours)	0.9326	0.9523	0.9188

**Table 3 sensors-19-01479-t003:** Comparison of different keypoints head on Parrot indoor dataset. (AP and AR represent the average accuracy and average recall, respectively. OKS indicates object keypoint similarity. For the explanations of each model, please refer to the text.)

**Model**	${\mathrm{AP}}_{\mathrm{OKS}=0.5:0.95}$	${\mathrm{AP}}_{\mathrm{OKS}=0.5}$	${\mathrm{AP}}_{\mathrm{OKS}=0.75}$
Lh-rcnn-k4	0.6453	0.8897	0.6811
Lh-rcnn-k8	0.6411	0.8804	0.6791
Lh-rcnn-k8-NL	0.6481	0.8759	0.6940
Lh-rcnn-k4-4	0.6891	0.9111	0.7446
Lh-rcnn-k4-4-RG (ours)	0.7415	0.9446	0.7908
**Model**	${\mathrm{AR}}_{\mathrm{OKS}=0.5:0.95}$	${\mathrm{AR}}_{\mathrm{OKS}=0.5}$	${\mathrm{AR}}_{\mathrm{OKS}=0.75}$
Lh-rcnn-k4	0.7523	0.9297	0.7985
Lh-rcnn-k8	0.7473	0.9154	0.7896
Lh-rcnn-k8-NL	0.7591	0.9213	0.8090
Lh-rcnn-k4-4	0.7764	0.9346	0.8275
Lh-rcnn-k4-4-RG (ours)	0.8054	0.9473	0.8479

**Table 4 sensors-19-01479-t004:** Comparison of different keypoints head on Parrot outdoor dataset. For the explanations of each model, please refer to the text.

**Model**	${\mathrm{AP}}_{\mathrm{OKS}=0.5:0.95}$	${\mathrm{AP}}_{\mathrm{OKS}=0.5}$	${\mathrm{AP}}_{\mathrm{OKS}=0.75}$
Lh-rcnn-k4	0.6124	0.8448	0.6571
Lh-rcnn-k8	0.6118	0.8492	0.6505
Lh-rcnn-k8-NL	0.6251	0.8509	0.6692
Lh-rcnn-k4-4	0.6743	0.8987	0.7205
Lh-rcnn-k4-4-RG (ours)	0.7298	0.9176	0.7754
**Model**	${\mathrm{AR}}_{\mathrm{OKS}=0.5:0.95}$	${\mathrm{AR}}_{\mathrm{OKS}=0.5}$	${\mathrm{AR}}_{\mathrm{OKS}=0.75}$
Lh-rcnn-k4	0.7084	0.8965	0.7631
Lh-rcnn-k8	0.7023	0.9005	0.7592
Lh-rcnn-k8-NL	0.7119	0.9056	0.7687
Lh-rcnn-k4-4	0.7686	0.9189	0.8049
Lh-rcnn-k4-4-RG (ours)	0.7869	0.9234	0.8195

**Table 5 sensors-19-01479-t005:** Comparisons of detection speed and accuracy on Parrot indoor dataset. Xception* is a small xception like the backbone shown in Table 1. CMU -Pose [59], G-RMI [35] and Mask-RCNN [42] are human keypoints detection algorithms; we used it by modifying the human keypoints to quadrotor keypoints.

Method	Backbone	Input Size	Speed (fps)	AP [0.5:0.95]
CMU-Pose [59]	VGG-19	654 × 368	20	0.5815
G-RMI [35]	Resnet50	1200 × 800	18	0.6446
Mask-RCNN [42]	Resnet50-FPN	1200 × 800	10	0.6672
Lh-rcnn-k4	xception*	1200 × 800	90	0.6453
Lh-rcnn-k8	xception*	1200 × 800	89	0.6411
Lh-rcnn-k8-NL [41]	xception*	1200 × 800	75	0.6481
Lh-rcnn-k4-4	xception*	1200 × 800	85	0.6891
Lh-rcnn-k4-4-RG (ours)	xception*	1200 × 800	71	0.7415

**Table 6 sensors-19-01479-t006:** The comparison of position and velocity tracking errors by Kalman filter with (NCA) or without (NCV) acceleration information.

	Position Tracking (m)		Velocity Tracking (m/s)	
	Mean Error	Standard Deviation	Mean Error	Standard Deviation
NCA	0.0757	0.1130	0.1228	0.2182
NCV	0.0934	0.1384	0.2034	0.3221

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
