# Peer review of "Drone Detection and Pose Estimation Using Relational Graph Networks"

_sensors, 2019, doi:10.3390/s19061479_

Round 1

Reviewer 1 Report

This work introduces a method for the detection and 6D pose estimation of a quadrotor drone, with the goal of using the attitude information from the pose to better predict its movement. The method uses a two-stage convolutional neural network to extract keypoints from the image, corresponding to the main elements of the drone structure, and later uses these keypoints to obtain the drone pose by solving a PnP problem. Results for several configurations of the system and state-of-the-art methods are presented, showing increasingly higher performance.

The manuscript presents some interesting results in line with the scope of the journal, but some changes must be done before it can be considered for publication:

Major modifications:

- Section 2.3: Much of what is explained here is directly extracted from [44]. The explanation must be reduced, or it must be clearly stated what is the novel contribution and what is included for the sake of completeness, with the appropriate citations.

- Section 3.4: I do not understand why the central value of matrix A is 0 and not the identity matrix, since this way it is not taking into account the previous velocity values. Moreover, the way to obtain the coefficients of matrix Q should be explained.

- The conclusions should be extended, including additional key figures from the previous section to support the claims quantitatively.

- English language and style should be revised throughout the entire manuscript.

Minor modifications:

- Ln 17: are emerged -> *have* emerged.
- Ln 22: During *the* past few years.
- Ln 84: The keypoints *that do not appear in the image or are obscured* will also be marked.
- Ln 90: Due to *the fact that* keypoint detection is based.
- Ln 100: Set A^l -> set A^l.
- Ln 109: as an input layer -> *be* an input layer.
- Figure 3: The figure should be explained better.
- Ln 174: *of* from the camera center -> from the camera center.
- Ln 183: I fail to understand this sentence, I think "singularity" should be replaced.
- Ln 197: The algorithm *is* as *follows*:
- Ln 218: convlutional -> convolutional.
- Ln 265: This should be connected to the previous paragraph.
- Ln 271: are donate -> denote.
- Ln 271: descripted -> described.
- Ln 277: The first part of the sentence seems incomplete.
- Ln 313: Both sigmas have the same subscript "w".
- Ln 318: (10)-(12) -> (12)-(14).
- Ln 326-329: Fix punctuation.
- Equation (21): I suggest to remove the asterisk, optionally including a middle dot, to denote multiplication.
- Ln 343: The figure number is missing.
- Section 3.4: Use consistent bold and italics for variables.
- Entire manuscript: *the* Figure -> Figure.
- Entire manuscript: Use consistent capitalization in section and subsection titles.

Author Response

We would like to express our sincere appreciation to the reviewers for their careful work in improving the presentation and quality of the paper. According to the reviewer's comments, we have carefully revised the paper and the modified parts are highlighted. We provide a cover letter to explain point-by-point the details of the revisions in the manuscript and our responses to the reviewers' comments. Due to space limitations, we put our responses in the attachment as a PDF file, thanks again for your comments.

Reviewer 2 Report

In this paper, the authors propose a novel relational graph network to inference quadrotor’s key points, and a improved PnP algorithm and filtering algorithm to estimate the 6DOF poses. The experiments seem to demonstrate the efficacy of the proposed method.

Overall, the paper is well organized and I am in favor of it. To make the paper more solid, here I have some feedbacks or suggestions for the authors to consider for revision.

-- Deep learning in the proposed method is to predict keypoints, and PnP method is used to estimate the 6DOF poses.  However, there exist the direct and end-to-end deep learning methods for 6DOF pose estimation. The authors should discuss the difference and relations between these different methods.

-- Regarding the experiment setting, the current experiment environment is indoor and the background is is very simple. What’s more, the ground is very similar to a checkerboard, which make the 6DOF pose even can be calculated by the traditional camera calibration methods.  I would like to suggest the authors to conduct another two groups of experiments. One is to make the indoor ground a little more complicated, and the other one is to change the indoor environment to the outdoor environment.

-- The current experiments lack the necessary baselines. The authors should compare the proposed method with the following deep learning method for pose estimation.

[1] A. Kendall, M. Grimes, and R. Cipolla. Posenet: A convolutional network for real-time 6-dof camera relocalization. In ICCV, pages 2938–2946, 2015.

[2] A. Kendall and R. Cipolla. Modelling uncertainty in deep learning for camera relocalization. In 2016 IEEE international conference on Robotics and Automation (ICRA), pages 4762–4769. IEEE, 2016.

[3] E. Brachmann, A. Krull, S. Nowozin, J. Shotton, F. Michel, S. Gumhold, and C. Rother. Dsac - differentiable ransac for camera localization. In The IEEE Conference on Computer Vision and Pattern Recognition (CVPR), July 2017.

[4] E. Brachmann and C. Rother. Learning less is more - 6d camera localization via 3d surface regression. In The IEEE Conference on Computer Vision and Pattern Recognition (CVPR), June 2018.

-- The current reference is inadequate. The authors should mention the deep learning methods for 6DOF pose estimation such as the abovementioned  [1]-[4].

-- Minor typo. “Figures ???” on Line 343 at Page 14.

Based on the above statement, I would like to let this paper go a round of major revision.

Author Response

We would like to express our sincere appreciation to the reviewers for their careful work in improving the presentation and quality of the paper. According to the reviewers' comments, we have carefully revised the paper and the modified parts are highlighted. We provide a cover letter to explain point-by-point the details of the revisions in the manuscript and our responses to the reviewers' comments. Due to space limitations, we put our responses in the attachment as a PDF file, thanks again for your comments.

Reviewer 3 Report

This paper presents a method for drone detection and pose estimation. While the method is interesting, in my opinion, there are several issues that must be completed or clarified.

First of all, in the abstract, the authors state that currently, 6D pose estimation algorithms depend on accurate pose annotation, and that their method overcomes this problem. However the experimental section shows that the database must be annotated manually. Please justify this.

In the introductory section, I have missed mentioning and commenting some works about UAVs pose estimation. Drone detection is sufficiently addressed, but no information is included about the state-of-the-art on UAV pose estimation.

In lines 57 and 58, the authors state that their experiments prove that directly using 2D human keypoints detection algorithm can not provide better results. Where in the experimental section have they compared their approach with any human keypoints detection algorithm?

In general, the figures where the quadrotor appears are quite difficult to see. Maybe making they larger would help to understand them. (figure 1, 6 and 9)

Do the authors process the images in any way to mitigate the effect of blur?

In figure 3(d), I guess that authors refer to frame t+1.

Again, about the necessary annotations of the real dataset, in line 254 the authors refer to the redundant annotations. What are they referring to with redundant annotations? Exactly, which are the manual annotations that are necessary?

In lines 272 to 274, the authors state that the keypoints detection performance is improved by dividing the keypoints into anchor and relational. However, have they develop any experiment to explicitly show it, along with relational graph block? 

As the authors state in lines 334-335, changing lighting conditions may have a negative impact upon the detection results. Have the authors explicitly tested this influence? 

Table 5 should include the units of the error and deviation. Also, in line 343, a non-existing figure is referenced.

Finally, the conclusion section should be extended. The authors must include a frank account of advantages and weaknesses of the method they propose. Also, they should describe future lines of improvement or completion of this framework.

Author Response

(The authors gave the same response as above.)

Round 2

Reviewer 1 Report

The authors have improved the paper and introduced all the modifications suggested in the comments. Thus, I recommend that the paper be published conditioned to the following minor modifications:

- Conclusion: By "key figures" I was referring to numerical values from the previous section. The authors should include these numerical performance indicators instead of references to figures.

- Ln 202: The sentence does not make sense in its current form either, but the problem is syntactic: in this context, "ambiguous" (adjective) should be used instead of "ambiguity" (noun).

p { margin-bottom: 0.25cm; line-height: 115%; }

Author Response

We would like to express our sincere appreciation to the reviewers for their careful work in improving the presentation and quality of the paper. According to the reviewers' comments, we have carefully revised the paper and checked English language and style. The modifications are as follows.

Point 1: Conclusion: By "key figures" I was referring to numerical values from the previous section. The authors should include these numerical performance indicators instead of references to figures.

Response 1: Thanks for your comments. I'm sorry to misunderstand your meaning.

We add the numerical performance indicators instead of references to figures: “Experiments and ablation studies (shown in Table 3) indicate that our relational graph network enhances 9.7% average precision compared to the baseline network in the same configuration. Besides, position and velocity errors (shown in Table 6) can be reduced by 19% and 40% respectively with the integration of acceleration estimation.”

The full conclusions are as follows:

In this paper, we propose a vision-based quadrotor pose and acceleration estimation method. Our method includes two parts, one is a novel relational graph network to improve the keypoints detection performance, the other is an improved PnP algorithm to acquire 6D quadrotor’s pose. These two algorithms are further integrated to estimate the acceleration and accurate velocity of the target. Experiments and ablation studies (shown in Table 3) indicate that our relational graph network enhances 9.7% average precision compared to the baseline network in the same configuration. Besides, position and velocity errors (shown in Table 6) can be reduced by 19% and 40% respectively with the integration of acceleration estimation.

The compatibility and performance of the algorithm is significantly adequate. It can achieve 71 fps on a desktop computer with an Nvidia Titan X GPU. Moreover, the algorithm is independent of the 3D model or accurate pose annotation for 6D pose estimation, meanwhile, it only requires the keypoints annotation of the target using captured images, and is especially suitable for non-cooperative target.

The downside of this algorithm is the accuracy of 6D pose decreases with the shrink of object size for small targets in the image. Therefore, in the future, we will improve the results on small target by employing adversarial training methods.

Point 2: Ln 202: The sentence does not make sense in its current form either, but the problem is syntactic: in this context, "ambiguous" (adjective) should be used instead of "ambiguity" (noun). 

Response 2: Thanks for your careful reading. We replace the word “ambiguity” to “ambiguous” in line 202. Thanks again for pointing out it.

Minor modifications:

Line 5, “the drone’s attitude information of the target becomes a key message” -> “the attitude information of the *target drone* becomes a key message”.

Line 6-7, “most of the object 6D pose estimation algorithms depending on” -> “most of the object 6D pose estimation algorithms *depend* on”.

Line 27, “only a small part *of* the field of view” -> “only a small part *in* the field of view”.

Line 47, “the other solution is first detect” -> “the other solution is first *to* detect”.

Line 52, “Recently, most studies used” -> “Recently, most studies *use*”.

Line 59, “the methods from Crivellaro et al. [28] were used.” -> “the *method* from Crivellaro et al. [28] *was* used.”

Line 70, “cannot get better results” -> “cannot *achieve* better results”.

Line 87-88, “Combined with keypoints detection results, *the* improved PnP algorithm and the filtering algorithm, *we* demonstrate that the method can actually reduce the UAVs’ state tracking error.”.

Line 92, “Since the order of *the* four motors” -> “Since the order of four motors”.

Line 99-100, “body bottom, we call them anchor keypoints.” -> “body bottom, *which are defined as* anchor keypoints.”.

Line 116, “Unlike them” -> “however, the difference is”.

Line 122, “the next layer features” -> “the next *layer’s* features”.

Line 124, “by residual connection” -> “by residual *connections*”

Line 151, “4 Channels” -> “4 *channels*”

Line 155, “RoIAlign is used for generating RoI boxes. Pooler resolution is set to 14” -> “RoIAlign is used *to generate* RoI boxes *and* pooler resolution is set to 14”

Line 157, “3.1” -> “section 3.1”

Line 175, “however” -> “besides”.

Line 217, “we compare its value with ground truth” -> “and its value is compared with ground truth”.

Line 225, “to cover prior drone shape and possible different sizes.” -> “to cover prior drone *shapes* and possible different sizes.”.

Line 244, “on a plane” -> “on the same plane”.

Line 255-256, “The other keypoints depend on the specific quadrotors, which do not contain enough common feauters. But these keypoints can be used to help locate the keypoints of the motors.” -> “The other keypoints which depend on the specific quadrotors without containing enough common feauters can be used to help locate the keypoints of the motors.”

Line 290, “which are used in training sets” -> “which are used *as* training sets”.

Line 355-466, “Our method includes two parts. One is a novel relational graph network to improve the keypoints detection performance. The other is an improved PnP algorithm to acquire 6D quadrotor’s pose.” -> “Our method includes two parts, *one* is a novel relational graph network to improve the keypoints detection performance, *the other* is an improved PnP algorithm to acquire 6D quadrotor’s pose.”

The modified parts are highlighted in the revised manuscript.

Reviewer 3 Report

The authors have addressed sufficiently the issues I raised in my review. In my opinion, the paper could be accepted for publication.

Author Response

We would like to express our sincere appreciation to the reviewers for their careful work in improving the presentation and quality of the paper. According to the reviewers' comments, we have carefully revised the paper and checked English language and style. The modifications are as follows.

Minor modifications:

Line 5, “the drone’s attitude information of the target becomes a key message” -> “the attitude information of the *target drone* becomes a key message”.

Line 6-7, “most of the object 6D pose estimation algorithms depending on” -> “most of the object 6D pose estimation algorithms *depend* on”.

Line 27, “only a small part *of* the field of view” -> “only a small part *in* the field of view”.

Line 47, “the other solution is first detect” -> “the other solution is first *to* detect”.

Line 52, “Recently, most studies used” -> “Recently, most studies *use*”.

Line 59, “the methods from Crivellaro et al. [28] were used.” -> “the *method* from Crivellaro et al. [28] *was* used.”

Line 70, “cannot get better results” -> “cannot *achieve* better results”.

Line 87-88, “Combined with keypoints detection results, *the* improved PnP algorithm and the filtering algorithm, *we* demonstrate that the method can actually reduce the UAVs’ state tracking error.”.

Line 92, “Since the order of *the* four motors” -> “Since the order of four motors”.

Line 99-100, “body bottom, we call them anchor keypoints.” -> “body bottom, *which are defined as* anchor keypoints.”.

Line 116, “Unlike them” -> “however, the difference is”.

Line 122, “the next layer features” -> “the next *layer’s* features”.

Line 124, “by residual connection” -> “by residual *connections*”

Line 151, “4 Channels” -> “4 *channels*”

Line 155, “RoIAlign is used for generating RoI boxes. Pooler resolution is set to 14” -> “RoIAlign is used *to generate* RoI boxes *and* pooler resolution is set to 14”

Line 157, “3.1” -> “section 3.1”

Line 175, “however” -> “besides”.

Line 217, “we compare its value with ground truth” -> “and its value is compared with ground truth”.

Line 225, “to cover prior drone shape and possible different sizes.” -> “to cover prior drone *shapes* and possible different sizes.”.

Line 244, “on a plane” -> “on the same plane”.

Line 255-256, “The other keypoints depend on the specific quadrotors, which do not contain enough common feauters. But these keypoints can be used to help locate the keypoints of the motors.” -> “The other keypoints which depend on the specific quadrotors without containing enough common feauters can be used to help locate the keypoints of the motors.”

Line 290, “which are used in training sets” -> “which are used *as* training sets”.

Line 355-466, “Our method includes two parts. One is a novel relational graph network to improve the keypoints detection performance. The other is an improved PnP algorithm to acquire 6D quadrotor’s pose.” -> “Our method includes two parts, *one* is a novel relational graph network to improve the keypoints detection performance, *the other* is an improved PnP algorithm to acquire 6D quadrotor’s pose.”

The modified parts are highlighted in the revised manuscript.